# Geomagnetic and visual cues guide seasonal migratory orientation in the nocturnal fall armyworm, the world's most invasive insect

Yi-Bo Ma[1†], Guijun Wan[1†], Yi Ji[1], Hui Chen[1,2], Bo-Ya Gao[1], Dai-Hong Yu[3], Eric Warrant[2], Yan Wu[4], Jason W Chapman[1,5], Gao Hu[1,4]*

[1]State Key Laboratory of Agricultural and Forestry Biosecurity, College of Plant Protection, Nanjing Agricultural University, Nanjing, China; [2]Lund Vision Group, Department of Biology, Lund University, Sölvegatan, Lund, Sweden; [3]Plant Protection Station of Yuanjiang County, Yuxi, China; [4]Key Laboratory of Surveillance and Management of Invasive Alien Species, Guizhou Education Department, Department of Biology and Engineering of Environment, Guiyang University, Guiyang, China; [5]Centre for Ecology and Conservation, University of Exeter, Penryn, United Kingdom

**\*For correspondence:**
hugao@njau.edu.cn

[†]These authors contributed equally to this work

**Competing interest:** The authors declare that no competing interests exist.

## eLife Assessment

This **fundamental** study presents experimental evidence on how geomagnetic and visual cues are integrated in a nocturnally migrating insect. The evidence supporting the conclusions is **compelling**. The work will be of broad interest to researchers studying animal migration and navigation.

**Abstract** The mechanisms guiding nocturnal insect migration remain poorly understood. Although many species are thought to use the geomagnetic field, the sensory basis of magnetic orientation in insects has yet to be clarified. We developed an indoor experimental system to investigate the integration of geomagnetic and visual cues in the seasonal orientation of a globally distributed pest moth, the fall armyworm (*Spodoptera frugiperda*), a highly invasive species which in the past decade has colonized almost all potentially habitable regions of the globe. Our results demonstrate that fall armyworms require both geomagnetic and visual cues for accurate migratory orientation, with visual cues being indispensable for magnetic orientation. When visual and geomagnetic cues are placed in conflict, moths become disoriented, although not immediately, indicating that sensory recognition of the conflict requires time to process. We also show that the absence of visual cues leads to a significant loss of flight stability, which likely explains the disruption in orientation. Our findings highlight that visual cues are critical for stable magnetic orientation in the fall armyworm, offering a basis for future investigations of visual-magnetic integration in noctuid migrants.

## Introduction

Numerous species of larger nocturnal moths, particularly in the family Noctuidae (hereafter 'noctuid moths'), undertake long-distance multigenerational migrations in the Northern Hemisphere. Every spring across North America and Eurasia, billions of noctuid moths move hundreds of kilometers northward to summer breeding grounds in the temperate zone, and in the subsequent fall, their progeny return southward to lower latitude wintering areas (*Holland et al., 2006*; *Satterfield et al.,*

*2020*; *Hu et al., 2016*; *Hu et al., 2025*; *Huang et al., 2024*). These migrations take place almost exclusively high above the ground, where fast winds facilitate rapid windborne transport leading to population redistribution over long distances (*Chapman et al., 2010*; *Chapman et al., 2015*; *Hu et al., 2016*; *Huang et al., 2024*). Some of the most abundant species involved in these migrations are the world's most destructive agricultural pests, and thus they are of huge importance for food security and economic prosperity (*Bauer and Hoye, 2014*; *Satterfield et al., 2020*; *Guo et al., 2020*; *Hu et al., 2025*). It is thus of paramount importance to understand all aspects of the migratory patterns of noctuid moths.

Tracking the seasonal progression of resources requires noctuid moths to move in the appropriate direction (northward in spring and southward in fall), a process entailing three linked steps. First, they determine the seasonally appropriate direction (north or south) in which to travel. Second, high-flying noctuid moths select transporting winds that are broadly aligned with this direction. Third, they adopt self-powered flight headings which are more or less aligned with the wind but, when required, offset to some degree to correct for crosswind drift (*Chapman et al., 2015*; *Hu et al., 2016*). Each step requires use of one or more compass senses based on globally stable cues to determine the required direction and behavioral responses (*Mouritsen, 2018*).

However, migration at night poses a considerable navigational challenge for noctuid moths and other night-flying insects (*Warrant and Dacke, 2011*; *Mouritsen, 2018*; *Foster et al., 2018*; *Gao et al., 2024*; *Grob et al., 2025*). This is because they cannot rely on the sun for compass information in the way that diurnally migrating butterflies and hoverflies do (*Mouritsen and Frost, 2002*; *Srygley and Dudley, 2008*; *Gao et al., 2020a*; *Massy et al., 2021*; *Pakhomov et al., 2025*). Despite this challenge, radar, tagging, and tethered-flight studies demonstrate that many larger nocturnal moths are highly sophisticated navigators, capable of selecting and maintaining appropriate movement directions with a high degree of accuracy, even when flying high above the ground (*Chapman et al., 2010*; *Dreyer et al., 2018b*; *Dreyer et al., 2018a*; *Menz et al., 2022*; *Chen et al., 2023*; *Dreyer et al., 2025*). Their ability to achieve seasonally appropriate migration directions typically surpasses that of diurnal windborne insect migrants (*Chapman et al., 2010*; *Hu et al., 2016*; *Werber et al., 2025*) and even matches the capability of nocturnal songbird migrants (*Alerstam et al., 2011*; *Chapman et al., 2016*). Clearly, migratory noctuid moths must possess one or more accurate compass senses, but in all species but one, the source of the compass information has yet to be elucidated (*Warrant and Dacke, 2011*; *Foster et al., 2018*; *Mouritsen, 2018*; *Freas and Spetch, 2023*; *Grob et al., 2025*).

The one exception to this lack of knowledge of the sensory basis of navigation in noctuid moths is the Bogong moth (*Agrotis infusa*) of Australia, which uses both a magnetic compass integrated with visual cues (*Dreyer et al., 2018b*) and a stellar compass (*Dreyer et al., 2025*) to migrate in seasonally appropriate directions. The Bogong moth is unique, however, among long-range migratory noctuids because a single generation makes bidirectional movements to and from a highly restricted geographic location in southeast Australia (*Warrant et al., 2016*), rather akin to the migration of eastern North American monarch butterflies to and from a restricted area of central Mexico (*Reppert et al., 2016*). This is distinct from most other noctuid moth migrants, whose multigenerational migrations involve back-and-forth movements between broad latitudinal zones (*Drake and Reynolds, 2012*; *Chapman et al., 2015*; *Gao et al., 2020b*; *Tong et al., 2022*; *Hu et al., 2025*) rather than to precise locations. Thus, other noctuid moths presumably do not require the same navigational precision and may therefore be expected to have simpler sensory capabilities than the Bogong moth. We examine this question using the fall armyworm (*Spodoptera frugiperda*), one of the world's most serious crop pests, as a model for understanding general noctuid moth migration and navigation capabilities.

The fall armyworm is a migratory crop pest native to the Americas. It breeds year-round in tropical regions and annually migrates into temperate regions of North America (*Nagoshi and Meagher, 2008*; *Westbrook et al., 2019*). Over the past decade, the species has invaded and rapidly spread across Africa, Asia, and Australasia (*Goergen et al., 2016*; *Kenis et al., 2023*; *Tay et al., 2023*), assisted by its high migratory capacity (*Chen et al., 2022*; *Chen et al., 2025*; *Gao et al., 2026*). It now causes huge yield losses in these regions. In Southeast Asia and South China, fall armyworm populations breed year-round in tropical regions. Each spring, they migrate northward as far as Northeast China, and their progeny return southward in the fall after summer breeding (*Li et al., 2020*; *Sun et al., 2021*; *Wu et al., 2021*; *Wu et al., 2022*).

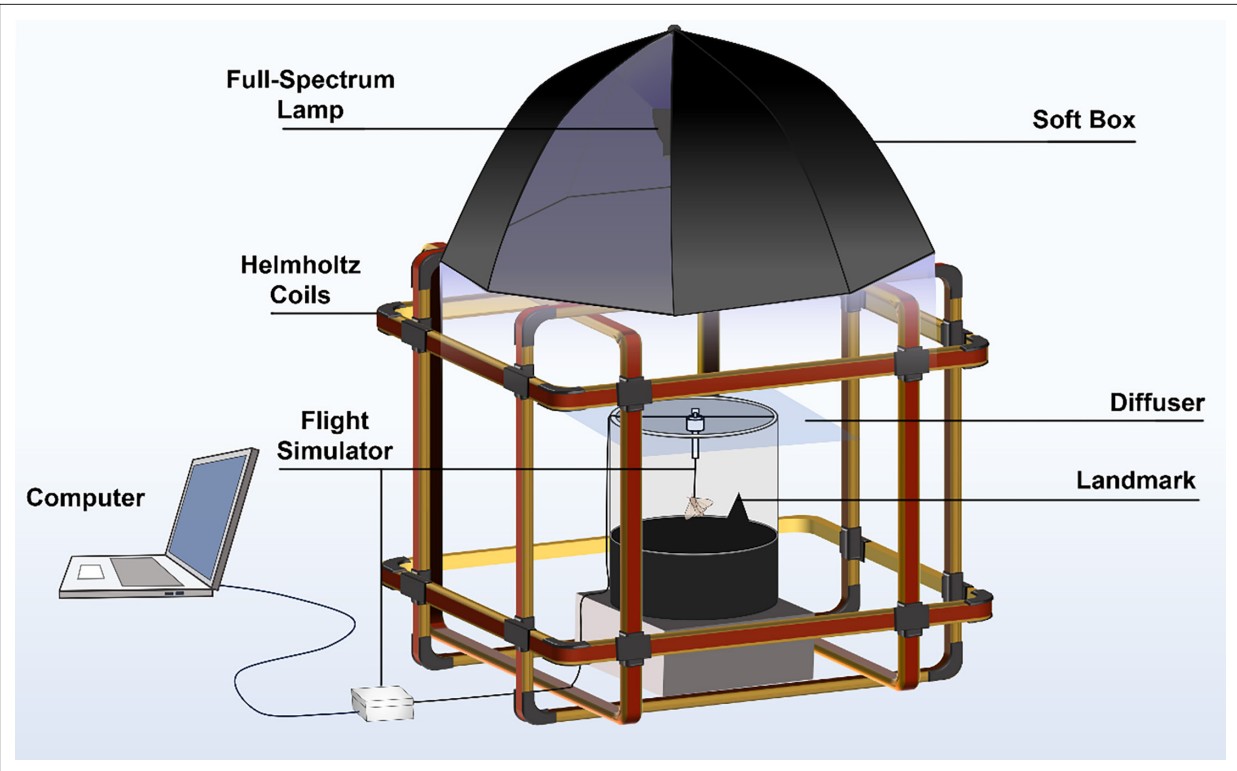

**Figure 1.** Schematic of the experimental setup for studying magnetic orientation in fall armyworms. Moths are tethered to a vertical shaft at the center of the virtual flight simulator, with an encoder recording their flight heading. A full-spectrum lamp illuminates the arena, while the computer controlling the experiment is positioned outside the light field to avoid interference. Moths are free to rotate in any direction during the assay. Full experimental details are given in Methods. The cylinder is illustrated as clear in the figure to reveal the internal setup, but it is opaque in the actual experiment.

We recently found that fall armyworms from the year-round range in Southwest China (Yunnan) exhibit seasonally appropriate migratory headings when flown outdoors in virtual flight simulators: heading northward in the spring and southward in the fall. This seasonal reversal is controlled by photoperiod (*Chen et al., 2023*). However, the compass mechanism that fall armyworms use to select seasonally appropriate headings remains unknown. The Earth's geomagnetic field is a ubiquitous and reliable source of compass information (*Mouritsen, 2018*), and thus might be expected to be the primary compass cue used by night-flying migratory moths. However, despite intriguing indications of magnetoreception in the oriental armyworm, *Mythimna separata* (*Xu et al., 2017*), so far only the highly specialized Bogong moth has been demonstrated behaviorally to use magnetic information integrated with visual cues for orientation (*Dreyer et al., 2018b*; *Dreyer et al., 2025*).

To test whether more generalist noctuid moth migrants also rely on a magnetic compass for orientation, we investigated an invasive fall armyworm population in China using a newly developed tethered-flight assay designed to quantify migratory orientation (*Figure 1*). While this indoor assay simplifies natural flight conditions, particularly due to the use of artificial visual cues, it provides a controlled framework for isolating the contributions of geomagnetic and visual cues. However, it should be noted that the setup does not include natural celestial cues, such as the moon, lunar polarization patterns, and stars, nor complex terrestrial topography, including mountains and water bodies, which could be incorporated in future studies to enhance ecological relevance. Despite these limitations, the current assay represents an important first step toward understanding how these signals function under more realistic ecological conditions. The results of these experiments are not only of fundamental interest to sensory biology (*Warrant and Dacke, 2011*), but also have applied importance, given that the fall armyworm is one of the most mobile and invasive crop pests in the world (*Kenis et al., 2023*; *Tay et al., 2023*).

## Results

### Integration of geomagnetic and visual cues in the seasonal migratory orientation in a field population of fall armyworms

Previous studies have demonstrated that fall armyworms exhibit distinct season-specific orientation preferences under natural field conditions; however, it is important to note that, in the absence of tracking experiments, the actual migratory destination regions remain unknown. To examine if fall armyworms integrate geomagnetic and visual cues for seasonal migratory orientation, we measured flight responses of tethered moths within a virtual flight simulator (*Dreyer et al., 2021*) using a modified experimental approach used in Bogong moth studies (*Dreyer et al., 2018b*; *Dreyer et al., 2025*). The simulator consisted of a polyvinyl chloride (PVC) cylinder incorporating a visual cue on the side (a black triangle rising above a black horizon; *Figure 1*). When tethered within the simulator, the moth is restrained but is free to rotate and thus take up any orientation it chooses. The simulator was placed within a 3D Helmholtz coil system. Under control conditions, the coil was switched off, leaving the Earth's natural magnetic field (NMF) unaltered; under these conditions, magnetic north and geographic north were closely aligned. However, when the coils were switched on, they reversed the horizontal direction of the local magnetic field by 180° (the 'changed magnetic field' [CMF]) relative to the NMF, while maintaining constant field intensity and inclination angle (*Figure 2—figure supplement 1*). Thus, the field direction within the experimental arena temporarily switches so that its 'local magnetic north' is aligned with geomagnetic south (GMS) and its 'local magnetic south' is aligned with geomagnetic north (GMN), but all other field parameters remain constant. The experimental setup consisted of recording moth flight headings across five consecutive 5 min phases (I, II, III, IV, V) of tethered flight under different experimental conditions that involved changing the azimuthal alignment of the visual cue and the local horizontal magnetic field component with respect to GMN and GMS (*Figure 2*, *Figure 2—figure supplement 1*). The assay was divided into these 5 min segments to provide the temporal resolution needed to detect changes in flight orientation as the relative alignment of magnetic and visual cues was systematically altered. In phases I and V, moths were exposed to the NMF, whereas in phases II–IV, moths were exposed to the CMF (*Figure 2*). During spring migration trials, the visual cue was aligned with GMN (the expected migratory orientation in spring) at the start of the trial, whereas in fall migration trials the setup was reversed, with the visual cue initially aligned with GMS (the expected migratory orientation in fall; *Figure 2*).

The first two experiments involved a field population of moths (collected as late-instar larvae) tested during spring and fall migration periods (*Figure 2A and B*). In phase I, the visual cue was aligned with the expected seasonal magnetic direction in the NMF, and moths exhibited significant group orientation toward the visual cue in both seasons (spring: mean vector [MV]=347.4° [95% CI: 319.5°, 16.71°], vector strength [$R^*$]=1.59, p<0.001, *Figure 2A-I*; fall: MV = 183.6° [95% CI: 211°,162°], $R^*$=1.76, p<0.001; *Figure 2B-I*). In phase II, the horizontal component of the geomagnetic field was rotated 180° (the CMF condition), creating a conflict between the visual cue direction and the expected magnetic orientation. Despite the shift in the magnetic field direction, moths continued to show significant group orientation toward the visual cue during this 5 min period (spring: MV = 319.1° [95% CI: 351°,283°], $R^*$=1.39, p<0.005, *Figure 2A-II*; fall: MV = 191.0° [95% CI: 235°,153°], $R^*$=1.16, p<0.05, *Figure 2B-II*), indicating that, at least initially, the visual cue was dominant compared to the magnetic cue. During phase III, which was a second 5 min period of the experimental conditions applied in phase II, moths lost their significant group-level orientation in both seasons (spring: $R^*$=0.67, p>0.05, *Figure 2A-III*; fall: $R^*$=0.34, p>0.05; *Figure 2B-III*), indicating that over time they had become confused due to the conflicting nature of the cues. Although we did not analyze flight activity level due to technical limitations, the loss of group orientation is unlikely to be attributable to a decline in the moths' motivation to maintain orientation during prolonged flight in the simplified experimental environment, as the Rayleigh test showed no significant differences in the $r$-values of individual flight vectors across phases I–III (*Figure 2A and B*; *Supplementary file 2*; *Supplementary file 3*). In phase IV, the visual cue was realigned with the expected seasonal magnetic orientation, but this time in the CMF (i.e. the cues were arranged in the same way as in phase I but rotated by 180°). Therefore, the moths should not be able to distinguish between phase I and phase IV and thus are expected to show the same orientation response in both phases. Indeed, the moths tended to show group orientation toward the congruent cues once again, albeit not quite reaching significance in spring (spring: MV = 245.2° [95% CI: 193.2°, 309.4°], $R^*$=0.95, 0.05<p<0.10, *Figure 2A-IV*; fall:

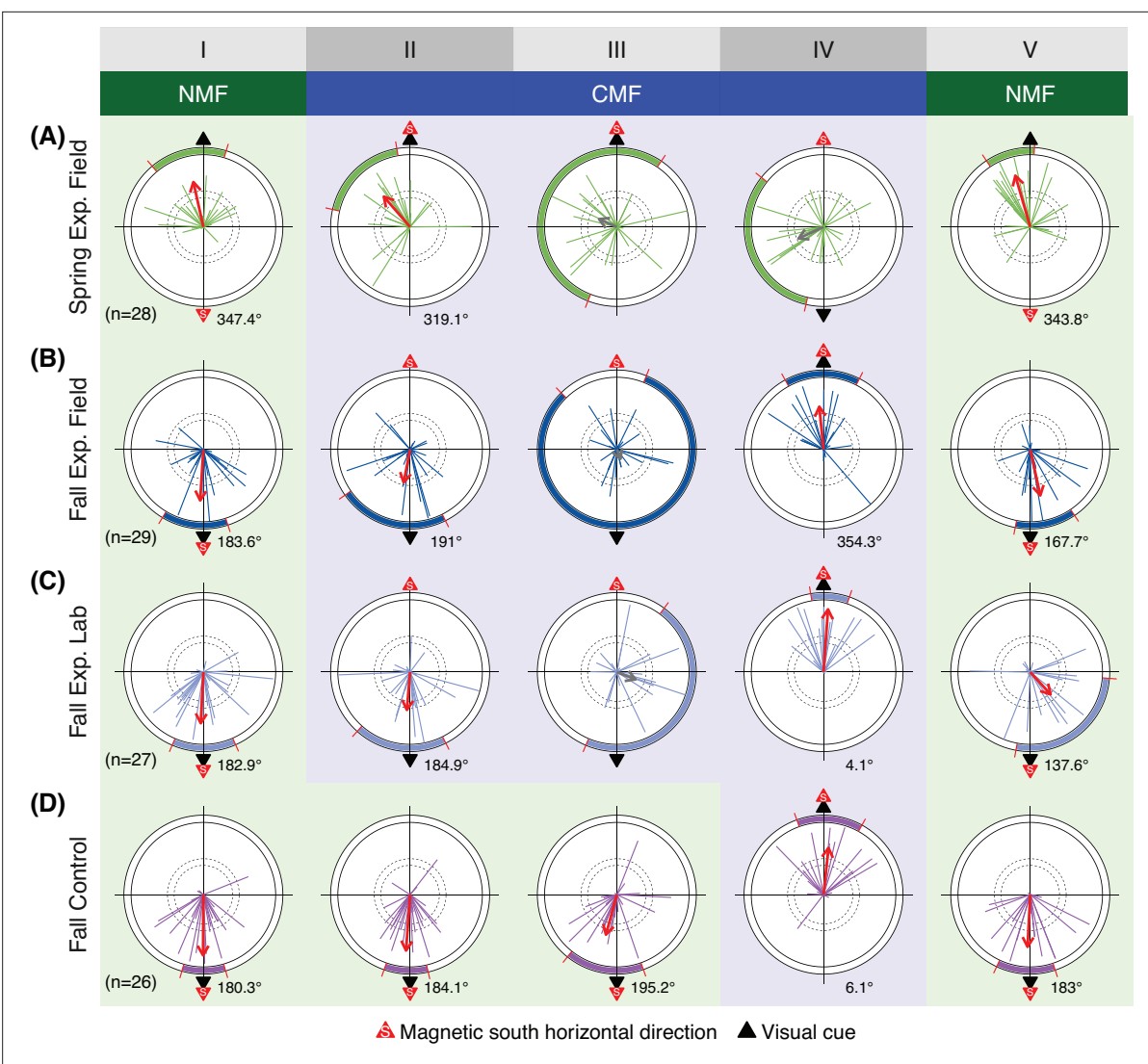

Figure 2. The Earth's magnetic field and visual cues guide migratory flight behavior in both a field population and lab-raised fall armyworms. (**A**) Flight orientation behavior of a spring field population of moths ('Spring Exp. Field') in response to visual and geomagnetic cues. (**B**) Flight orientation behavior of a fall field population of moths ('Fall Exp. Field') in response to visual and geomagnetic cues. (**C**) Flight orientation behavior of lab-raised fall-conditioned moths ('Fall Exp. Lab') in response to visual and geomagnetic cues. (**D**) Flight orientation behavior of lab-raised control fall-conditioned moths ('Fall Control'), tested with consistent visual and geomagnetic alignment. For simplicity and consistency, we tested only conditions in which the visual cue pointed in the putative migratory direction, coinciding with magnetic north in spring and with magnetic south in fall. The putative migratory direction was defined based on the seasonally different orientation directions revealed by our previous field experiments (*Chen et al., 2023*). Each group underwent five sequential 5 min phases (I–V), with each subplot representing individual moths' flight directions in the simulator. The length of each vector represents individual directedness (*r*), ranging from 0 to 1, where the outer edge of the plot corresponds to *r*=1. The thick mean vector (MV) arrow represents the weighted average of individual orientations, calculated using Moore's modified Rayleigh test (see Methods), and is red when there is significant group orientation but gray when it is not significant. The *R\** value quantifies the directedness of the MV. Dashed circles indicate thresholds for statistical significance, with radii corresponding to p<0.05 and p<0.01. Shaded sections of the outer diameters of the circles represent the 95% confidence limits of the group orientation. The outermost radius represents *R\**=2.5. The black triangle denotes the position of the visual cue, while the red triangle indicates the direction of the expected migratory orientation in each season (north in spring, south in fall). The experimental setup included both the natural magnetic field (NMF, panels with light green background) and a changed magnetic field (CMF, panels with light blue background) where the horizontal magnetic field direction was reversed; further details of the magnetic field parameters in the NMF and CMF are shown in *Figure 2—figure supplement 1*. The Fall Exp. Field is the only experiment for which results are based on pooled data from 2023 and 2024, with year-specific results provided in *Figure 2—figure supplement 2*.

The online version of this article includes the following source data and figure supplement(s) for figure 2:

**Source data 1.** Results of rayleigh tests for each individual.

**Figure supplement 1.** The magnetic field conditions during experimental procedures in 2023 and 2024.

*Figure 2 continued on next page*

*Figure 2 continued*

**Figure supplement 1—source data 1.** Frequency spectrum of magnetic field (B) at 10 kHz RBW.

**Figure supplement 2.** Year-specific analysis of orientation behavior in field-captured armyworms during the fall migration season (2023 and 2024).

**Figure supplement 2—source data 1.** Results of rayleigh tests for each individual in fall field experiments (2023–2024).

**Figure supplement 3.** Spectral distribution of light provided by the full-spectrum lamp.

**Figure supplement 3—source data 1.** Spectral distribution of light provided by the full-spectrum lamp.

MV = 354.3° [95% CI: 331.4°, 27.2°], $R^*$=1.49, p<0.005; *Figure 2B-IV*). Finally, in phase V, the initial configuration was restored, and moths regained significant group orientation toward the congruent visual cue and the expected magnetic orientation in the NMF (spring: MV = 343.8° [95% CI: 326.56°, 2.92°], $R^*$=1.84, p<0.001, *Figure 2A-V*; fall: MV = 167.7° [95% CI: 191°, 146°], $R^*$=1.62, p<0.001, *Figure 2B-V*). These results demonstrate that fall armyworm integrates geomagnetic field information with visual cues to achieve stable orientation. However, when geomagnetic and visual cues do not align with expected seasonal directions, moths gradually lose orientation, reinforcing the critical role of cue integration in maintaining migratory stability.

## Integration of geomagnetic and visual cues is consistent in laboratory-reared fall armyworms

Our previous research showed that fall armyworms reared under artificially simulated fall migratory season and tested indoors with optic flow and starry sky projections exhibited southward orientation in a flight simulator, consistent with the behavior observed in wild individuals tested outdoors under natural night skies (*Chen et al., 2023*). To determine whether lab-raised moths exposed to simulated seasonal photoperiods respond the same way to geomagnetic and visual cues, we tested a population reared under a simulated fall photoperiod in the same experimental setup as that of the field population fall group (*Figure 2C*). The results closely matched those from the field population (*Figure 2B*). In phase I, with visual cues aligned to GMS in the NMF, lab-raised moths oriented significantly southward (MV = 182.9° [95% CI: 203°, 156°], $R^*$=1.77, p<0.001, *Figure 2C-I*). When the geomagnetic field was reversed (CMF), moths initially oriented toward the visual cue, showing no immediate response to the shifted field (MV = 184.9° [95% CI: 222°, 153°], $R^*$=1.31, p<0.05, *Figure 2C-II*). After an additional 5 min period without changes, significant group-level orientation was lost ($R^*$=0.69, p>0.05, *Figure 2C-III*). When the visual cue was realigned with the changed GMS, significant orientation was restored (MV = 4.1° [95% CI: 350.7°, 19.3°], $R^*$=2.17, p<0.001, *Figure 2C-IV*), though the direction was opposite that in phase I. Finally, returning to the original setup restored the initial orientation pattern (MV = 137.6° [95% CI: 189.9°, 94.9°], $R^*$=1.04, p<0.05, *Figure 2C-V*). These results demonstrate that lab-raised fall armyworms integrate geomagnetic and visual cues similarly to field populations, emphasizing the role of photoperiod during development in shaping seasonal multimodal migratory orientation.

To confirm that the loss of orientation in experimental groups was due to the cue conflict rather than diminished motivation to maintain orientation arising from a prolonged test duration, we conducted a control experiment with the lab-raised simulated fall population, where the visual cue and GMS remained consistently aligned. In this group, moths maintained significant orientation in their expected migratory direction throughout the experiment when exposed to congruently aligned cues, irrespective of whether under NMF or CMF conditions (*Figure 2D*; directional statistics in *Supplementary file 1*). This confirms that the loss of orientation observed in experimental groups was driven by misalignment between visual and geomagnetic cues.

## Fall armyworms show a delayed response to changes in magnetic fields

Our results show that migratory fall armyworms rely on both the Earth's magnetic field and visual cues to determine their flight direction. However, for moths to maintain constant orientation, the two sensory inputs must remain congruent, and when they come into conflict, the moths become disoriented at the group level (but not at the individual level; *Figure 2*). Notably, this response to the conflict between visual and geomagnetic cues was not instantaneous but delayed until the second 5 min experimental phase with this condition, consistent with similar results from Bogong moths (*Dreyer et al., 2018b*). Initial analyses using a 5 min resolution provided limited temporal detail.

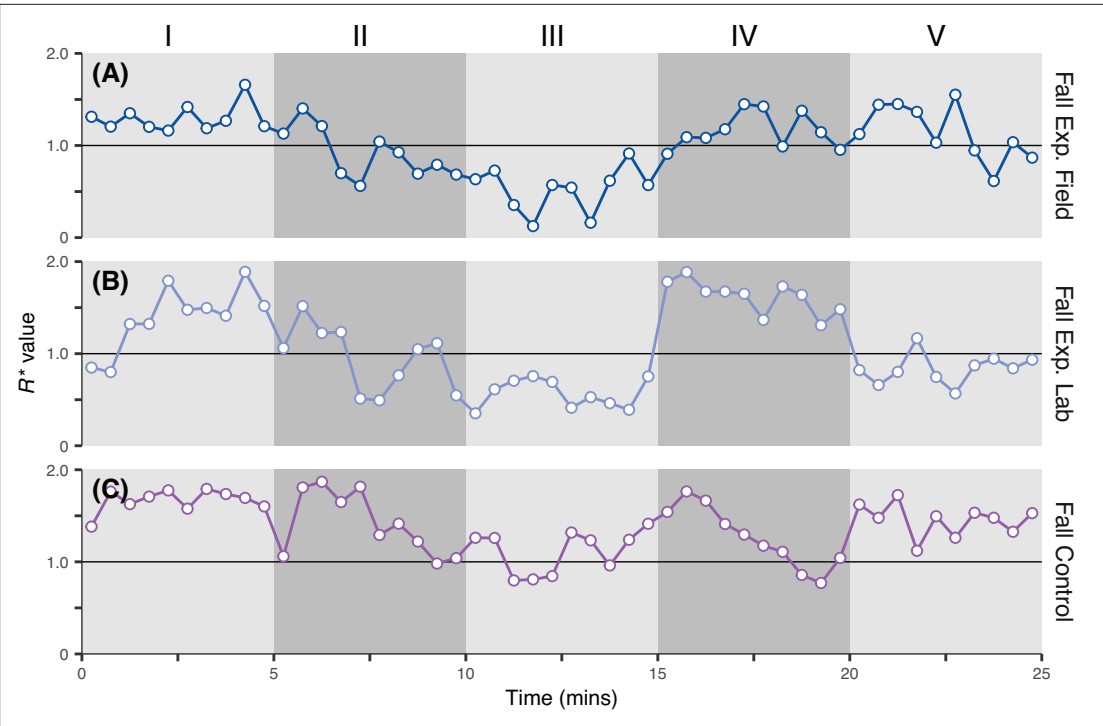

**Figure 3.** Fall armyworms exhibit a delayed response to changes in magnetic-visual cue alignment. The behavioral data for (**A**) fall field population experimental ('Fall Exp. Field', data from *Figure 2B*), (**B**) fall lab-raised experimental ('Fall Exp. Lab', data from *Figure 2C*), and (**C**) fall lab-raised control ('Fall Control', data from *Figure 2D*) groups were analyzed in 30 s time bins, resulting in 10 bins over each 5 min phase of the experiment. For each group, Moore's modified Rayleigh test was applied, and the obtained $R^*$ values were plotted against time. $R^*>1$ indicates a significant collective orientation within that 30 s interval, while $R^*<1$ indicates the absence of significant group-level orientation.

The online version of this article includes the following source data for figure 3:

**Source data 1.** Time-binned analysis of collective orientation using Moore's modified rayleigh test ($R^*$).

To improve resolution, we subdivided each 5 min phase into ten 30 s intervals and independently analyzed directional consistency using Moore's Rayleigh test to calculate the $R^*$ for each 30 s period (*Figure 3*).

During phase I, $R^*$ values in both experimental groups remained >1 (*Figure 3A-I and B-I*), indicating significant orientation behavior. When the magnetic field was rotated to conflict with visual cue directions, $R^*$ values did not drop immediately but instead declined gradually over time (*Figure 3A-II and B-II*). By phase III, $R^*$ values were consistently <1, indicating a loss of significant orientation (*Figure 3A–III and B-III*). Upon realigning the magnetic field with visual cues, $R^*$ values showed a gradual increase, eventually exceeding 1 in both groups (*Figure 3A-IV and B-IV*). When the original configuration was restored, the field population group consistently exhibited $R^*$ values above 1 (*Figure 3A-V*), while the lab-raised group had $R^*$ values hovering around 1 (*Figure 3B-V*). These results indicate that fall armyworms require time to resolve conflicting navigational inputs before losing group orientation entirely or to react to congruent cues and regain group orientation. In the lab-raised control group, where the geomagnetic field and visual cues remained consistently aligned, $R^*$ values consistently remained above 1 (*Figure 3C*), confirming that the decline in group-level orientation observed in the experimental groups was driven by conflicts between geomagnetic and visual cues, mirroring the results found in Bogong moths (*Dreyer et al., 2018b*).

## Visual cues are essential for magnetic orientation in fall armyworms

Growing evidence supports the idea that geomagnetic cues provide essential compass information for orientation across a broad range of taxa, supporting diverse navigational tasks (*Mouritsen, 2018*; *Dreyer et al., 2018b*; *Fleischmann et al., 2018*; *Wan et al., 2021*; *Grob et al., 2024*; *Goforth et al., 2025*). To determine whether geomagnetic input alone is sufficient for flight orientation in fall

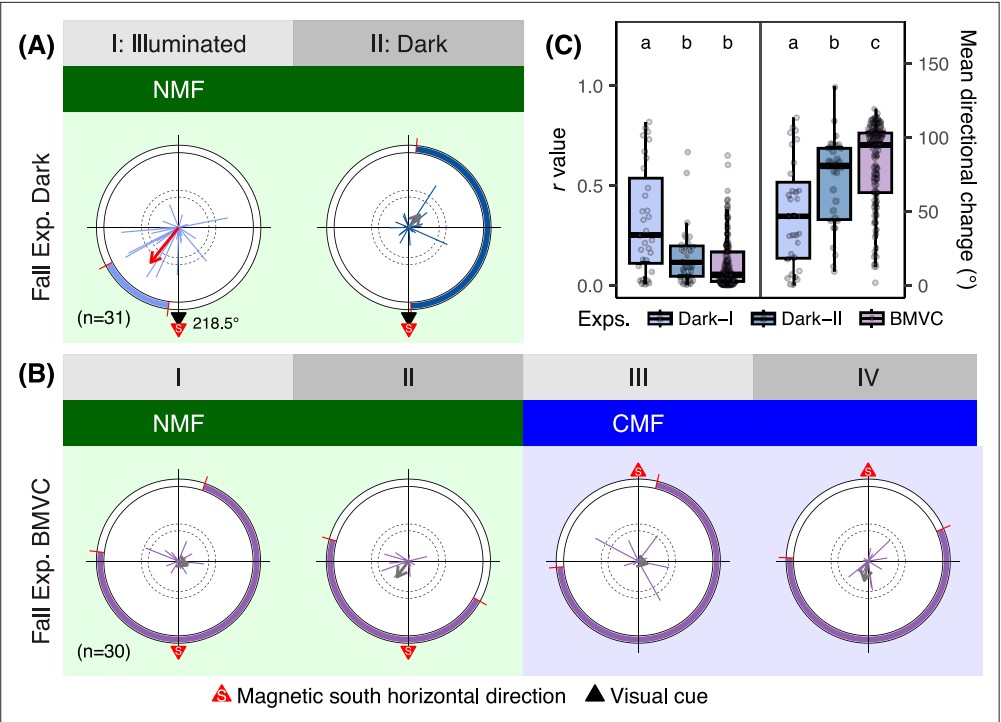

**Figure 4.** Visual information is essential for maintaining group flight orientation in fall armyworms. (**A**) The lab-raised, fall-conditioned population lost significant group orientation under complete darkness ('Fall Exp. Dark'). (**B**) The lab-raised, fall-conditioned population exhibited a significant loss of group orientation under illuminated conditions where visual cues were reduced to the bare minimum ('Fall Exp. BMVC'), i.e., obvious visual cues such as the black triangle shown in **Figure 1** were not provided; however, because the experiments were conducted under illuminated conditions, complete elimination of all visual information is impossible. (**C**) The distribution of Rayleigh test *r*-values for individual moth orientations across different treatment groups (*left*) and the distribution of average directional change per second (*right*), the latter reflecting flight stability. The box plots represent the interquartile range (IQR), with the horizontal line inside indicating the median. Whiskers extend to the most extreme data points within 1.5 times the IQR. Pairwise comparisons of the *r*-values were performed using the Wilcoxon rank-sum test and Wilcoxon signed-rank test. (The Wilcoxon signed-rank test is only used to compare non-independent [i.e. paired or related] samples. For more information, please refer to the Statistical analysis section.) Multiple comparisons were corrected by the Benjamini-Hochberg method. Detailed statistical results are provided in **Supplementary file 4**. Comparisons of the average directional change per second were performed using the same method as that used for *r*-values, with statistical results provided in **Supplementary file 5**. Groups labeled with different letters differ significantly (p<0.05). We also analyzed the *r*-values and average directional change per second in the Fall Exp. Field, Fall Exp. Lab, Exp. Fall Control, and the experiment shown in this figure, with results consistent with those shown here (see **Supplementary files 4 and 5**, **Figure 4—figure supplement 1**).

The online version of this article includes the following source data and figure supplement(s) for figure 4:

**Source data 1.** Individual rayleigh test data from Fall Exp. Dark and Fall Exp. BMVC.

**Source data 2.** Individual r values and mean directional change in Fall Exp. Dark and Fall Exp. BMVC.

**Figure supplement 1.** The Lack of Visual Cues and Other Necessary Visual Information Significantly Affects Moth Orientation Ability and Flight Stability.

**Figure supplement 1—source data 1.** Rayleigh test r values and mean directional change for individual moth orientations across treatment groups.

armyworms, we conducted a series of experiments to evaluate the role of visual cues in their orientation behavior. In particular, we examined whether these moths could maintain orientation in the absence of visual cues.

As a control, we first tested lab-raised, fall-conditioned moths in an arena where visual landmarks were aligned with GMS in the NMF (**Figure 4A**). When illuminated, moths showed significant group orientation toward the visual cue (MV = 218.48° [95% CI: 242°, 187°], *R*\*=1.50, p<0.005, **Figure 4A-I**),

consistent with our previous results (*Figure 2B*). To determine whether moths could maintain orientation without visual information, we then removed all light sources, allowing moths to continue flying in total darkness. Under these conditions, moths lost significant group-level orientation ($R$*=0.5, p>0.05, *Figure 4A-II*), suggesting that their orientation behavior is dependent on the presence of light. The loss of orientation may be due to the absence of usable visual landmarks or the lack of light in a dark environment causing the moths to lose their sense of direction. However, it is still unclear how the absence of usable visual landmarks in non-dark environments would affect the moths' orientation behavior.

To further investigate this, we tested a lab-raised, fall-conditioned population of moths in a simulator, primarily to determine whether moths could maintain orientation under illuminated conditions where visual cues on the interior walls of the cylindrical flight arena were minimized. In this experiment, we provided a uniform visual environment (i.e. under illuminated conditions, visual cues were not provided on the interior walls of the cylindrical flight arena). After recording 10 min of orientation behavior under the NMF, we reversed the horizontal component of the geomagnetic field using Helmholtz coils (CMF) and recorded their flight orientation for an additional 10 min. Despite the altered magnetic field direction and sufficient time for adaptation, moths exhibited no significant group orientation across all four testing phases (*Figure 4B*; directional statistics in *Supplementary file 1*). The lack of group-level orientation under near-uniform visual conditions suggests two possibilities: individual moths may have maintained stable flight, but due to the absence of reliable orientation cues, their group-level directional choices were random; or their flight may have been unstable, making it difficult to assess individual orientation behavior. We first compared the differences in the Rayleigh test $r$-values of moths across different experimental phases (*Figure 4C*, *left*), using the Wilcoxon signed-rank test for the non-independent Fall Exp. and Fall Dark groups, and the Wilcoxon rank-sum test for all other independent group comparisons. This is because the Rayleigh test evaluates the concentration of data, with an $r$-value close to 1 indicating highly consistent flight directions, and close to 0 suggesting more random flight directions. The results showed that under conditions with visual cues (Fall Exp. Dark-I), moths had significantly higher $r$-values compared to conditions of complete darkness (Fall Exp. Dark-II) or visual cues reduced to the bare minimum (Fall Exp. BMVC).

We also considered the limitations of the Rayleigh test, as its $r$-value only reflects the overall directional tendency and does not reveal real-time flight stability. For example, a moth exhibiting two stable flight phases, one oriented north and the other south, would yield a low $r$-value despite maintaining stability within each phase. Conversely, a moth losing flight stability could also produce a low $r$-value, making it difficult to distinguish between these two scenarios based on $r$-value alone. Therefore, we introduced the analysis of per-second angular change (*Figure 4C*, *right*) to assess dynamic fluctuations in flight behavior, providing a clearer understanding of whether the observed differences in $r$-values were due to stable orientation or a loss of flight control. Our analysis showed that the experimental groups tested under conditions lacking visual cues exhibited significantly lower flight stability than the other experimental groups (*Figure 4C*; *Figure 4—figure supplement 1*). This conclusion is similar to the earlier analysis based on individual Rayleigh test $r$-values (*Figure 4C*). Moths in these conditions exhibited significantly higher angular change rates, suggesting that the reduced stability likely contributed to the lower directedness, as indicated by the reduced vector lengths ($r$). Notably, even though a small subset of individuals in these groups managed to sustain stable flight, their directional choices appeared random rather than goal-oriented (*Figure 4A-II and B*).

## Discussion

The integration of geomagnetic and visual cues for flight orientation has so far been experimentally demonstrated only in the Australian Bogong moth, a species undertaking a precisely oriented, long-distance round-trip migration to and from a highly restricted aestivation site within a single generation (*Dreyer et al., 2018b*). Although this life-history strategy is uncommon among migratory insects, our development of an indoor behavioral paradigm in the fall armyworm, a globally distributed nocturnal migratory insect with multi-generational and partially migratory patterns in China, extends this dual-cue orientation strategy to a species exhibiting a migratory ecology more typical of insects in general. These findings demonstrate that this orientation mechanism is not unique to the Bogong moth and may instead represent a conserved orientation mechanism broadly employed across migratory moth species, though additional species will need to be evaluated to demonstrate this. Crucially,

we show that visual cues are indispensable for magnetic orientation in the fall armyworm, as directional responses disappeared in the absence of structured visual input. Furthermore, our findings reinforce recent results on seasonal shifts in orientation behavior in this species (*Chen et al., 2023*), highlighting the ecological relevance of this guidance system.

Our findings thus emphasize the importance of integrating multiple cues for successful orientation. It is often assumed that animals lack magnetic orientation capabilities if they fail to orient under changes in the geomagnetic field (*Pakhomov et al., 2025*), either in appropriate lighting conditions or in darkness, without additional cues. However, our study demonstrates that the absence of group seasonal orientation under geomagnetic fields, in full-spectrum light or in darkness, does not preclude magnetic orientation when additional sensory cues are integrated. Specifically, in the Northern Hemisphere, a magnetic compass provides global directional cues for fall armyworms, enabling them to align magnetic north (or magnetic pole) and south (or magnetic equator) with visual cues during spring and fall, respectively, to facilitate seasonal orientation—a hallmark of annual long-distance migration in migratory species. Although the magnetic compass is critical, as evidenced by the loss of group orientation when the horizontal geomagnetic field was reversed while intensity and inclination remained unchanged, our results reveal that magnetic orientation cannot occur without appropriate visual cues. We also show that the loss of flight stability observed when visual cues are minimal is likely a major cause of the lack of group magnetic orientation. This highlights the complexity of magnetic orientation strategies in nocturnal migratory insects, emphasizing the crucial interaction between the magnetic compass and visual navigation systems. Further investigation by altering the geomagnetic polarity and vertical geomagnetic component is required to determine whether the magnetic compass guiding seasonal orientation in nocturnal migratory fall armyworms is polarity-sensitive, inclination-sensitive, or both – a factor which is pivotal to understanding the biophysical mechanisms underlying magnetoreception (*Mouritsen, 2018*; *Hore and Mouritsen, 2016*; *Grob et al., 2024*).

Desert ants and Bogong moths have also been shown to integrate geomagnetic and visual cues for navigation. Desert ants rely on a magnetic compass during look-back-to-the-nest behavior, combining magnetic cues with local visual landmarks, such as their nest (*Grob et al., 2024*). In contrast, Bogong moths have been demonstrated to integrate global stellar cues with the Earth's magnetic field for orientation (*Dreyer et al., 2025*). Like Bogong moths, fall armyworms can migrate hundreds, or even thousands, of kilometers over several successive nights (*Warrant et al., 2016*; *Li et al., 2020*). Recent work on skyglow has shown that large-scale nocturnal luminance gradients can bias the orientation of fall armyworm moths, with individuals tending to orient toward darker regions of the sky (*Ji et al., 2025*). However, whether fall armyworms are capable of detecting and exploiting naturally occurring light-dark gradients, such as those generated by the Milky Way, for orientation under natural conditions remains unknown.

Under natural migratory conditions, fall armyworms are unlikely to experience a visual environment dominated by only a single, static, and highly salient landmark. Celestial cues are intermittently available, and local features shift during flight. Given this complexity, several key questions about orientation in fall armyworms remain unresolved. (a) One important issue is the role of celestial cues when local landmarks are absent. Experimental disruptions to local features could test whether celestial cues, such as those from the moon or stars, can compensate for the absence of local landmarks. (b) When multiple visual cues are available, it is essential to understand how they are weighted and integrated with geomagnetic information. To explore this, experiments could vary the presence and prominence of different visual cues (e.g. celestial cues vs. local landmarks) and examine how they interact with geomagnetic information to guide orientation. (c) Transient visual cues, such as shifting clouds or changing lighting conditions, may also influence orientation. Experiments could simulate these transient cues and assess how they affect orientation, both in isolation and in conjunction with geomagnetic cues. (d) Finally, given that luminance gradients are unlikely to be absent in natural nocturnal environments, how such visual cues may be integrated with the geomagnetic field for orientation remains a critical topic for future research. To investigate this, studies could manipulate luminance gradient strength and stability and test how these gradients interact with geomagnetic cues to influence orientation, further advancing our understanding of the integration of visual and geomagnetic cues during nocturnal migration.

In conclusion, our findings underscore the indispensability of visual cues in facilitating seasonal magnetic orientation. To date, comparable results have been demonstrated only in another nocturnal

moth, the Bogong moth (*Dreyer et al., 2018b*). Unlike many vertebrate studies, in which behavioral changes can be induced by magnetic field manipulations alone under simplified conditions (*Mouritsen, 2018*; *Burda et al., 1990*), these findings highlight the complex, multimodal nature of navigation in nocturnal migratory insects, as well as the methodological considerations associated with studying their orientation behavior. This work advances our understanding of migratory orientation strategies and raises the possibility that similar magnetic-visual integration mechanisms may be shared among long-distance migratory moths. Furthermore, the application of genome editing in fall armyworms, combined with the indoor behavioral paradigm developed here, provides a promising avenue for dissecting the molecular basis of magnetoreception and the interplay between sensory systems.

## Methods

### Field population

All wild populations used in this study were collected from farmland in Yuanjiang Hani and Yi Autonomous County, Yunnan Province, a region primarily characterized by maize cultivation. Fall-generation populations were collected during September to October of 2023 and 2024, while spring-generation populations were collected from April to May in 2024, as mature larvae at the 5th to 6th instar stages. Each larva was individually collected and reared in a cylindrical container, fed with fresh maize leaves from the collection site until pupation. Upon pupation, larvae were transferred to transparent, thin plastic cups sealed with plastic film. A cotton ball moistened with water was placed in each cup to maintain adequate humidity during the pupal stage. After eclosion, adults were individually provided with a 10% honey solution. All experimental individuals were reared at the Yuanjiang Plant Protection Station, a facility located near agricultural fields and well isolated from urban light pollution. To maintain natural environmental conditions, all electrical devices were strictly excluded from the facility, and windows were kept open around the clock to allow for natural light cycles and ambient temperature conditions.

### Lab-raised population

The laboratory-reared population of the fall armyworm used in this study was primarily derived from wild individuals collected from a maize field in Yuanjiang, Yunnan Province (Google Maps coordinates: 23.604°N, 101.977°E) during October to November 2022, with the exception of individuals used in experiments Fall Exp. Dark and Fall Exp. BMVC (see *Figure 4*), which were collected from the same field location during October to November 2024. All experiments were conducted using individuals from the 3rd to 5th laboratory-reared generations. The Fall Exp. lab and fall control populations were reared from January to May 2023, whereas the Fall Exp. Dark and Fall Exp. BMVC populations were reared from January to February 2024.

For behavioral assays conducted under ambient temperature conditions, a programmed photoperiodic regimen was implemented to simulate the seasonal changes in day length typical of autumn in East Asia. All treatments were maintained at a constant temperature of 27 ± 1°C and a relative humidity of 60% ± 5%. In the fall photoperiod treatment, the initial light cycle at egg hatching was set to 13 hr of light and 11 hr of darkness (13 L:11D), followed by a daily reduction of 2 min in the light period. By the time adult moths emerged and were subjected to experimental testing approximately 1 month later, the photoperiod had been adjusted to approximately 12L:12D.

### Attachment of tethering stalks on moths

We used the same method for the attachment of tethering stalks on moths as in the previous study on fall armyworm moth orientation (*Chen et al., 2023*). Prior to tethering, unmated 2-day-old adult moths were housed in plastic cups and sedated at 4°C for at least 1 min to facilitate handling. The moths were then transferred to an operational platform, where scales were carefully removed from the junction of the dorsal thorax and abdomen to ensure secure attachment. The tether consisted of a slender, non-magnetic copper stalk measuring 0.75 mm in diameter and approximately 2 cm in length. The terminal end of the stalk was bent into a fork-like structure, which was carefully affixed to the prepared thoracic-abdominal junction using Pattex PSK12CT-2 glue. To maintain consistency, this standardized tethering method was applied uniformly across all experimental protocols.

## Behavioral apparatus

Our flight simulator system is consistent with the one used in previous research on fall armyworm (*Chen et al., 2023*), utilizing the Flash flight simulator system. This system was developed based on the early design of the Mouritsen-Frost flight simulator, was developed with a more user-friendly human-machine interaction interface and an integrated optic-flow system (Flash Flight Simulator Data Acquisition System.exe is available on Mendeley Data: 10.17632/6jkvpybswd.1), and adapted for our experiments conducted in Yuanjiang (longitude 101.98°E, latitude 23.60°N). However, in contrast to previous studies, we did not include optic flow in order to avoid introducing additional visual complexity. The moths are able to freely rotate on the horizontal plane within the simulator, with their azimuth measured by an encoder system with a resolution of 0.9°. The flight direction (relative to magnetic north) is recorded in real time by the Flash flight simulator data acquisition system developed by Hui Chen, and the data is saved as angle values in a text file. In our experiments, the encoder (made of non-magnetic materials) samples the azimuth five times per second and is equipped with a graphical interface, which allows for continuous monitoring of the moth's azimuth and flight status to reconstruct the flight path. The encoder is connected to the axis that holds the moth, with the axis measuring 15 cm in length and 1 mm in diameter, made of carbon fiber material and wrapped in a carbon fiber tube to protect its structure.

The arena consists of an opaque PVC cylinder with a diameter of 400 mm, a height of 500 mm, and a thickness of 5 mm as the main structure, surrounded by white wallpaper, with a 20 cm high black wallpaper strip at the bottom simulating the horizon. A visual cue is provided by a black isosceles triangle (10 cm high, 10 cm base) made from black wallpaper and fixed to the horizon at the bottom of the arena. The entire arena is placed on a square wooden board with a side length of 50 cm and a thickness of 1 cm, covered with black blackout cloth. The bottom of the wooden board is equipped with a plastic base to reduce friction, ensuring that the arena can rotate smoothly and quickly during the experiment, changing the direction of the visual cues while avoiding interference with the moth's activity.

In the BMVC experiment, we used an opaque white acrylic cylinder with the same dimensions as the PVC cylinder as the arena. No modifications were made to the interior of the arena, primarily because the interior of the white acrylic cylinder is smooth and uniform enough, and any additional treatment could potentially introduce noticeable visual cues. We also removed the carbon fiber crossbar used to fix the simulator and instead fixed it directly to the top acrylic cover to further minimize potential visual cues that the moth might use. Our goal was to minimize the visual cues available to the moth as much as possible, rather than attempting to completely eliminate all visual cues.

A DH1766-series linear, three-channel, programmable DC power supply (Beijing Dahua Radio Instrument Factory, China) with adjustable current and voltage was used to power the full-spectrum LED employed in the experiments. The light source was enclosed within a 90 cm diameter GODOX softbox to ensure even distribution of light within the experimental arena. The softbox is fixed to a Helmholtz coil, with the flight simulator located at its center. The top of the arena is equipped with a UV-transmitting acrylic panel and covered with Lee Filters 250 semi-white diffusion paper to control the amount of light transmitted. Spectral measurements, which include both spectral irradiance and composition, were taken at the position corresponding to the moth's head using an ATP2000P spectrometer (Optosky Technology Co. Ltd., Xiamen, China), ensuring consistent lighting conditions across all visual treatments (see *Figure 2—figure supplement 3*). The ambient light level in the experimental environment was measured to be below 1 lux using a Testo 540 lux meter (Testo SE & Co. KGaA, Titisee-Neustadt, Germany). Further work is still required to compare the illuminance used in this study with that under natural conditions, which are inherently variable.

## Artificial simulation of the Earth's magnetic field

A three-dimensional pair of Helmholtz coils (Nanjing Science Sky Technology Limited) was used to manipulate the Earth's magnetic field. The coils measured 1200×1100×1000 mm$^3$, with a cold-state direct current resistance of 2 Ω per coil and an insulation resistance exceeding 10 MΩ. The magnetic field strength at the central X, Y, and Z axes exceeded 10 GS@6A (maximum current), with a spatial uniformity of 0.5% within a φ144×144×144 mm$^3$ volume. At the start of each experiment, we used a TM4300B handheld three-axis fluxgate magnetometer to measure the ambient magnetic field strength and spatial vector intensity at the experimental site (averaged over three measurements). A DH1766

series linear three-channel output precision programmable DC power supply (Beijing Dahua Radio Instrument Factory, China) with adjustable current and voltage was employed to apply a controlled current to the 3D Helmholtz coils, reversing the horizontal geomagnetic component by approximately 180°. Crucially, we ensured that total magnetic field intensity (E), horizontal component intensity (H), and magnetic inclination angle ($\alpha$) remained statistically unchanged between the changed magnetic field (CMF) and the natural magnetic field (NMF) (see *Figure 2—figure supplement 1A–D*). Electromagnetic noise at the experimental site was quantified by measuring the magnetic component of the time-dependent electromagnetic field across a 10 kHz resolution bandwidth. Measurements were performed using a Spectran NF-5035 spectrum analyzer (1 Hz to 30 MHz, Aaronia AG) connected to an MDF-9400 magnetic field antenna (9 kHz to 400 MHz, Aaronia AG) via a UBBV-MDF960X preamplifier (9 kHz to 60 MHz, 25 dB gain) and a 10 m RF cable. The frequency range from 1 to 10 MHz was measured using the external antenna-preamplifier configuration, while low-frequency noise between 2 kHz and 1 MHz was measured using the internal magnetic probe of the spectrum analyzer. During each measurement session, the spectrum analyzer continuously scanned the frequency range for 40 min, and values were logged using the instrument's peak-hold mode. Pronounced noise peaks were detected in the 2 kHz to 1 MHz range, whereas noise levels from 1 to 10 MHz were low and relatively stable, ranging between 0.01 and 0.001 nT with only minor fluctuations (see *Figure 2—figure supplement 1E*).

## Experimental procedure

The orientation experiment followed the protocols established in our previous study (*Chen et al., 2023*). Experiments began after complete darkness in the night sky: the field population was tested after adapting to the natural dark environment for at least 60 min, while the lab-raised population started after adapting to the dark environment for at least 60 min following the light-to-dark cycle switch in the incubator, in a laboratory away from large electrical equipment, artificial noise, plants, and other potential sources of interference. The computer screen was positioned away from and facing away from the experimental setup. Weak red light was used only when attaching the tethering stalks and fixing the moth to the simulator. After the moth was fixed, a waiting period of 20–30 s was allowed for the subject to stabilize before data recording began. During this process, it was verified whether the insect could switch between clockwise and counterclockwise rotations. If the insect rotated in only one direction and could not fly stably, no further experiment was conducted. The encoder's horizontal placement was also checked to ensure it was not affected by tilting. The moth's wing vibrations were confirmed to be strong, with equal amplitude on both wings (indicating that the contact glue did not interfere with the wings). During the experiment, the insect's flight state was monitored by observing fluctuations in the pointer values on the experimental software. If the values did not change within 10 s, the experimenter needed to approach the flight arena and listen for any sounds of wing vibrations. If the insect stopped flying, the experimenter gently tapped the wall to stimulate the insect to continue flying. We analyzed individuals that were able to maintain fewer than four instances of wingbeats cessation during the experiment.

## Statistical analysis

All statistical analyses and graph generation were performed using R (version 4.1.3), accessible at https://www.r-project.org/. A custom R script, incorporating bootstrap confidence intervals, was used for Moore's modified Rayleigh test (MMRT) – a non-parametric test increasingly applied in orientation behavior studies (*Massy et al., 2021*; *Chen et al., 2023*), the script is available on: 10.1098/rspb.2021.1805. MMRT accounts for both mean directions and directedness (vector lengths) of individuals, making it an alternative to the Rayleigh test. MMRT assesses whether a group exhibits significant directional tendencies, using the parameter $R^*$ (a scaled alternative to $r$ in circular regression), which quantifies the strength of orientation within a group, and the MV, which represents the rank-weighted mean direction of the group. Polar plots display individual vectors, derived from azimuth data collected during the experiment by the encoder at a sampling rate of five times per second, and subsequently analyzed using the Rayleigh test. Each vector representing an individual moth's mean flight direction and directedness ($r$-value). The vector length reflects the proportion of time a moth maintains a particular direction, ranging from $r=0$ (completely un-oriented) to $r=1$ (fully oriented).

MMRT requires the analysis of each individual's average direction and $r$-value to determine the group-level mean MV. It is important to note that the magnitude of $R*$ is dependent on sample size.

To evaluate the directional orientation of moths in different treatment groups, we performed Rayleigh tests on individual flight directions. Subsequently, we used the Friedman test to examine whether different experimental phases within each experiment influenced the $r$-values of moth orientation, as this test is appropriate for comparing repeated measurements from the same individuals (*Supplementary file 2*). In the initial test, only the Exp. Spring experiment showed significant differences. Therefore, we further compared the flight stability and $r$-values among its different experimental phases (*Supplementary file 3*). Because the $r$-value exhibited non-normal distributions, the Wilcoxon signed-rank test was employed. Since these comparisons involved repeated measurements from the same individuals, pairwise comparisons were conducted. Because multiple comparisons were performed, failing to apply a correction would increase the likelihood of false positives; therefore, the Benjamini-Hochberg procedure was used to control the false discovery rate (FDR).

The Rayleigh $r$-values for each individual were also used to compare orientation strength across experiment. We accounted for the non-independence between Exp. Dark I and Exp. Dark II by using the paired Wilcoxon signed-rank test for their pairwise comparison. All other comparisons among independent experiments were performed using the Wilcoxon rank-sum test. The multiple comparisons were corrected using the Benjamini-Hochberg procedure to control the FDR. Detailed statistical results are provided in *Supplementary file 4*.

To evaluate flight stability, we measured the angular difference between consecutive seconds of recorded flight behavior and calculated the average directional change per second. This metric reflects the consistency and stability of individual flight orientation. Differences among treatment groups were assessed using the Wilcoxon signed-rank test (*Supplementary file 5*) because some of the data did not follow a normal distribution. We also accounted for the non-independence between the Exp. Dark I and Exp. Dark II experiments, for which paired comparisons were conducted. Multiple comparisons were corrected using the Benjamini–Hochberg procedure to control the FDR.

## Acknowledgements

This work was supported by the National Key Research and Development Program of China (2021YFD1400700), the Fundamental Research Funds for the Central Universities (KJJQ2025013, RENCAI2025031), the Joint Research Program of State Key Laboratory of Agricultural and Forestry Biosecurity (SKLJRP2507), and the National Natural Science Foundation of China to (32202289).

## Additional information

### Funding

| Funder | Grant reference number | Author |
|---|---|---|
| National Key Research and Development Program of China | 2021YFD1400700 | Gao Hu |
| Fundamental Research Funds for Central Universities | KJJQ2025013 | Gao Hu |
| Fundamental Research Funds for Central Universities | RENCAI2025031 | Jason W Chapman |
| Joint Research Program of State Key Laboratory of Agricultural and Forestry Biosecurity | SKLJRP2507 | Gao Hu |
| National Natural Science Foundation of China | 32202289 | Bo-Ya Gao |

| Funder | Grant reference number | Author |
| --- | --- | --- |

The funders had no role in study design, data collection and interpretation, or the decision to submit the work for publication.

## Author contributions
Yi-Bo Ma, Software, Formal analysis, Investigation, Visualization, Methodology, Writing – original draft, Writing – review and editing; Guijun Wan, Conceptualization, Data curation, Supervision, Writing – original draft, Writing – review and editing; Yi Ji, Data curation, Investigation; Hui Chen, Software, Methodology; Bo-Ya Gao, Data curation, Investigation, Writing – review and editing; Dai-Hong Yu, Resources, Investigation; Eric Warrant, Jason W Chapman, Supervision, Writing – review and editing; Yan Wu, Methodology; Gao Hu, Conceptualization, Supervision, Funding acquisition, Visualization, Writing – review and editing

## Author ORCIDs
Yi-Bo Ma 
Guijun Wan 
Eric Warrant 
Jason W Chapman 
Gao Hu 

None https://doi.org/10.7554/eLife.109098.4.sa1
Reviewer #2 (Public review): https://doi.org/10.7554/eLife.109098.4.sa2
Author response https://doi.org/10.7554/eLife.109098.4.sa3

# Additional files

## Supplementary files
Supplementary file 1. Detailed data of flight orientation behavior assay.

Supplementary file 2. Friedman test results of Rayleigh's r values across experimental groups.

Supplementary file 3. Pairwise comparison results of the mean Rayleigh's r values of Exp. Spring across different experimental phases, using the Wilcoxon signed-rank test with multiple-comparison correction performed by the Benjamini-Hochberg method.

Supplementary file 4. Multiple comparisons of mean Rayleigh test r values across experimental conditions using the Wilcoxon rank-sum test (adjusted by the Benjamini–Hochberg method). Note: A: Fall Exp. Field; B: Fall Exp. Lab; C: Fall Control; D: Fall Exp. Dark I; E: Fall Exp. Dark II; F: Fall Exp. BMVC.

Supplementary file 5. Wilcoxon rank-sum test for mean angular velocity (directional change per second) across experimental conditions (Benjamini–Hochberg correction). Note: A: Fall Exp. Field; B: Fall Exp. Lab; C: Fall Control; D: Fall Exp. Dark-I; E: Fall Exp. Dark-II; F: Fall Exp. BMVC.

## Data availability
All data are available in the main text or the supplementary materials.

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
