## [Editor Report · eLife Assessment]

This **fundamental** study presents experimental evidence on how geomagnetic and visual cues are integrated in a nocturnally migrating insect. The evidence supporting the conclusions is **compelling**. The work will be of broad interest to researchers studying animal migration and navigation.

---

## [Referee Report · None]

[Editors' note: this version has been assessed by the Reviewing Editor without further input from the original reviewers. The authors have addressed the comments raised in the previous round of review.]

**Reviewer #1 (Public review):**

Summary:

The manuscript by Ma et al. provides robust and novel evidence that the noctuid moth Spodoptera frugiperda (Fall Armyworm) possesses a complex compass mechanism for seasonal migration that integrates visual horizon cues with Earth's magnetic field (likely its horizontal component). This is an important and timely study: apart from the Bogong moth, no other nocturnal Lepidoptera has yet been shown to rely on such a dual-compass system. The research therefore expands our understanding of magnetic orientation in insects with both theoretical (evolution and sensory biology) and applied (agricultural pest management, a new model of magnetoreception) significance.

The study uses state-of-the-art methods and presents convincing behavioural evidence for a multimodal compass. It also establishes the Fall Armyworm as a tractable new insect model for exploring the sensory mechanisms of magnetoreception, given the experimental challenges of working with migratory birds. Overall, the experiments are well designed, the analyses are appropriate, and the conclusions are generally well supported by the data.

Strengths:

• Novelty and significance: First strong demonstration of a magnetic-visual compass in a globally relevant migratory moth species, extending previous findings from the Bogong moth and opening new research avenues in comparative magnetoreception.

• Methodological robustness: Use of validated and sophisticated behavioural paradigms and magnetic manipulations consistent with best practices in the field. The use of 5 min bins to study a dynamic nature of magnetic compass which is anchored to a visual cue but updated with latency of several minutes is an important finding and a new methodological aspect in insect orientation studies.

• Clarity of experimental logic: The cue-conflict and visual cue manipulations are conceptually sound and capable of addressing clear mechanistic questions.

• Ecological and applied relevance: Results have implications for understanding migration in an invasive agricultural pest with expanding global range.

• Potential model system: Provides a new, experimentally accessible species for dissecting the sensory and neural bases of magnetic orientation.

Weaknesses:

Overall, this is a strong study, and the authors have completed an excellent major revision.

---

## [Referee Report · Reviewer #2 (Public review)]

Summary:

The work titled "Geomagnetic and visual cues guide seasonal migratory orientation in the nocturnal fall armyworm, the world's most invasive insect" provided experimental evidence on how geomagnetic and visual cues are integrated, and visual cues are indispensable for magnetic orientation in the nocturnal fall armyworm.

Strengths:

It has been demonstrated that the Australian Bogon moth could integrate global stellar cues with the geomagnetic field for long distance navigation. However, data are lacking for other insects. This study suggested that the integration of geomagnetic and visual cues may represent a conserved navigational mechanism broadly employed across migratory insects.

Weaknesses:

The visual cues used in the indoor experimental system designed by the authors may have some limitations in ecological relevance. The author may need more explanations on this experimental system.

In the revised manuscript, the authors have added explanations in the discussion section. I am fine with the revision.

---

## [Author Response]

The following is the authors’ response to the previous reviews

**Public Reviews:**

**Reviewer # 1 (Public review):**
(1) Structure and Presentation of Results• I recommend reordering the visual-cue experiments to progress from simpler conditions (no cues) to more complex ones (cue-conflict). This would improve narrative logic and accessibility for non-specialist readers. The authors have chosen not to implement this suggestion, which I respect, but my recommendation stands.

Thank you for this suggestion. We understand your point that presenting the experiments from simpler to more complex conditions may seem more intuitive. However, we have kept the original order because it better reflects the logic of the study itself. Our work first asked whether fall armyworms, like the Bogong moth, use a magnetic compass that is integrated with visual cues. Only after establishing this behavioral feature did we go on to test whether visual cues are required to maintain magnetic orientation. To make this reasoning clearer to readers, we have explicitly stated in the Introduction that magnetic orientation in the Bogong moth depends on the integration of visual cues, which provides clearer context for the experimental design.

(2) Ecological Interpretation• The authors should expand their discussion on how the highly simplified, static cue setup translates to natural migratory conditions, where landmarks are dynamic, transient, or absent. Specifically, further consideration is needed on how the compass might function when landmarks shift position, become obscured, or are replaced by celestial cues. Additionally, the discussion would benefit from a more consolidated section with concrete suggestions for future experiments involving transient, multiple, or more naturalistic visual cues. This point was addressed partially in one paragraph of the Discussion, which reads as follows:"In nature, they are likely to encounter a range of luminance-gradient visual cues, including relatively stable celestial cues as well as transient or shifting local features encountered en route. Although such natural cues differ from our simplified laboratory stimulus, they may represent intermittently sampled visual inputs that can be optimally integrated with magnetic information, with the congruency between visual and magnetic cues likely playing a key role in maintaining a stable compass response. Whether the cues are static or changing, brief periods without them may still allow the subsequent recovery of a stable long-distance orientation strategy. Determining which types of natural visual cues support the magnetic-visual compass, and how they interact with magnetic information, including how their momentary alignment or angular relationship is integrated and how such visual cue-magnetic field interactions may require time to influence orientation, together with elucidating the genetic and ecological bases of multimodal orientation, will be important objectives for future research." While this paragraph is informative, the wording remains lengthy, somewhat unclear, and vague. Shorter, clearer statements would improve readability and impact. For example:• How could moths maintain direction during periods when only the magnetic field is present and visual landmarks are absent?• Could celestial cues (e.g., stars) compensate, and what happens if these are also obscured?• What role does saliency play when multiple visual landmarks are present simultaneously?• How might a complex skyline without salient landmarks affect orientation?Including simple, concise sentences that pose concrete open questions and suggest experimental designs would strengthen the discussion without creating space issues. In my view, a comprehensive discussion of how the simplified, static cue setup relates to natural migratory conditions-where landmarks are dynamic, transient, or absent-would add significant value to the paper.

Thank you for this constructive and insightful comment. You correctly point out that our articulation of the ecological relevance of the simplified, static cue setup was not sufficiently clear. We also agree that the original wording in the Discussion remained overly general. In the revised Discussion, we updated the manuscript to incorporate recently published findings on the use of light–dark gradients for orientation in fall armyworms. However, we explicitly note that it remains unclear whether fall armyworms can exploit naturally occurring luminance gradients, such as those generated by the moon, for orientation under natural conditions. We further emphasize that during natural migration the visual environment is dynamic, with celestial cues available intermittently and local visual features changing continuously during flight. In this context, we outline several key unresolved questions, including whether celestial cues can compensate when local landmarks are absent; how multiple visual cues are weighted and integrated with geomagnetic information; how transient visual cues (like moving clouds or changing illumination) influence orientation; and how luminance gradients that are common in natural nocturnal environments interact with the geomagnetic field to support orientation. For each of these issues, we briefly suggest experimental approaches to guide future research.

(3) Methodological Details and Reproducibility• The lack of luminance level measurements should be explicitly highlighted.

Thank you for your helpful suggestion. You are right that luminance level is an important experimental parameter. We have stated this information in the Methods section under Behavioral apparatus: “The ambient light level in the experimental environment was measured to be below 1 lux using a Testo 540 lux meter (Testo SE & Co. KGaA, Titisee-Neustadt, Germany). Further work is still required to compare the illuminance used in this study with that under natural conditions, which are inherently variable.” This point is also clarified in the legend of Figure S3 in the supplementary material.

• The authors chose not to adjust figure legends by replacing "magnetic South" with "magnetic North." While I believe this would be more conventional and preferable, this is ultimately a minor stylistic issue.

Thank you very much for your suggestion. We understand your point and agree that using “magnetic North” would be more conventional. However, because our experiments focus on the orientation behavior of the autumn population, magnetic South is aligned with the landmark direction representing the potential migratory direction, which we believe makes the figures more intuitive for readers. We therefore consider this a minor stylistic issue.

(4) Conceptual Framing and Discussion• Although the authors made a good attempt to explain the limitations of using an artificial visual cue, I believe there is room or a more explicit argument. For example, it could be stated clearly that this species is unlikely to encounter a situation in nature where a single, highly salient landmark coincides with its migratory direction. Therefore, how these findings translate to real migratory contexts remains an open question. A sentence or two making this point directly would strengthen the discussion.

Thank you for your helpful suggestion. We now address this point explicitly in the Discussion, noting that fall armyworms are unlikely to experience a natural visual environment dominated by a single, static, and highly salient landmark coinciding with their migratory direction. Consequently, how these findings translate to real migratory contexts remains an open question.

(5) Technical and Open-Science Points• Sharing the R code openly (e.g., via GitHub) should be seriously considered. The code does not need to be perfectly formatted, but making it available would be highly beneficial from an open-science perspective.

Thank you for the suggestion. We agree that making code openly available is valuable from an open-science perspective. The MMRT script used in this study is Moore’s Modified Rayleigh Test, available from the original publication by Massy et al. (2021; https://doi.org/10.1098/rspb.2021.1805). In the previous version, we only cited this reference in the Materials and Methods section; we have now added a direct link to the script to improve clarity and accessibility. We have also provided a public link to the data-recording scripts used in the Flash Flight Simulator (https://doi.org/10.17632/6jkvpybswd.1). This repository additionally includes a map-based optical flow script that was not used in the present study but is shared for completeness.

**Reviewer #1 (Recommendations for the authors):**
• LL. 133-137 (end of paragraph starting with "The fall armyworm is a migratory crop pest native to the Americas"): Suggest splitting into shorter, clearer sentences. The limitations of this method could be better articulated here and elaborated in the Discussion.

Thank you for this suggestion. We have revised this paragraph by splitting it into shorter, clearer sentences and by articulating the limitations of this method more explicitly. These limitations are further elaborated in the Discussion.

• LL. 181-185 (end of paragraph starting with "To examine if fall armyworms integrate geomagnetic and visual cues for seasonal migratory orientation"): It would be helpful to state explicitly that season-specific headings have been confirmed in the lab using a flight simulator, but destination regions remain unknown without further tracking experiments.

Thank you for this helpful suggestion. We have now clarified in the revised manuscript that season-specific orientation headings have been confirmed in the laboratory using a flight simulator, while the actual migratory destination regions remain unclear in the absence of tracking experiments.

• LL. 230-234 (start of paragraph "Our previous research showed that fall armyworms reared under artificially simulated fall conditions…"): Clarify which migratory season is being referenced.

Thank you for this helpful suggestion. We have clarified in the text that the migratory season referenced here is the autumn migratory season. In addition, we have added information in the Methods to specify the actual calendar season during which the insects were reared under the simulated conditions.

• LL. 270-272 (middle of Fig. 2 caption): Suggest explicitly mentioning that for this population, the seasonally appropriate direction is southbound in autumn and northbound in spring, as this may not be clear to non-specialists.

Thank you for this helpful suggestion. We have now explicitly stated the seasonally appropriate migratory directions for this population, indicating southbound migration in autumn and northbound migration in spring, to improve clarity for non-specialist readers.

• LL. 421 (middle of paragraph starting with "We also considered the limitations of the Rayleigh test…"): Add that the groups lacking visual cues exhibited "lower directedness as per lower vector length (r)" in addition to lower flight stability.

Thank you for this helpful suggestion. We further note that the conclusions drawn from the flight stability analysis are consistent with those based on individual r-value analyses.

• LL. 499-501 ("unlike some vertebrates that can rely solely on magnetic information (Mouritsen, 2018)"): This point is slightly downplayed. It should be emphasized that nearly all tested vertebrates and invertebrates (e.g., birds, mole rats, fish, frogs, and other insects) demonstrate a magnetic compass without requiring visual landmarks. Moths are the only tested invertebrates so far that show landmark-magnetic field dependency for their magnetic compass to be manifested in a behavioural orientation response in Flight Simulator.

Thank you for this important comment. We agree that this point represents a key synthesis in the Discussion, as it concerns how our findings relate to, and differ from, magnetic orientation demonstrated in other animal groups. We have therefore expanded the Discussion to note that studies have shown that some animals can exhibit directional preferences in simplified visual environments solely in response to changes in the magnetic field, and we now cite representative examples from birds and mole rats. At the same time, we also acknowledge important methodological and phenotypic differences among taxa. In particular, moths’ magnetic orientation has been assessed using a flight simulator, a setup in which stable directional behavior must be actively maintained during continuous movement. This is an important difference from orientation assays in birds during take-off or in terrestrial mammals such as mole rats. Moreover, whether birds and other animals rely on visual input to detect or calibrate magnetic information under certain conditions remains an open question. We therefore emphasize here both the phenotypic differences observed across experimental systems and the methodological considerations.

• LL. 560-565 (paragraph starting with "Our flight simulator system (Dreyer et al., 2021) …"): Suggest clarifying what the Flash flight simulator system is and how it differs from the Mouritsen-Frost flight simulator.

Thank you for this suggestion. We have added a brief clarification of the Flash flight simulator and how it differs from the Mouritsen–Frost system.

• LL. 605-608 ("Spectral measurements …"): Explicitly mention that total illuminance was not measured and that further work is required to compare the illuminance used with natural conditions which of course vary.

Thank you for this helpful suggestion. We agree that total illuminance is an important factor. We have now added a statement noting that the ambient light level in the experimental environment was measured to be below 1 lux using a Testo 540 lux meter, and we further acknowledge that additional work is required to compare the illuminance used in this study with that under naturally variable conditions.

• LL. 628-641 (end of paragraph starting with "Electromagnetic noise at the experimental site ... "): Explain why this matters for interpreting behavioural responses. Highlight that although conditions were somewhat magnetically noisy which based on the past work may disrupt magnetic compass as it was shown in birds (eg Engels et al. 2014 Nature), the observed magnetic response under certain conditions indicates that the magnetic sense remained functional when landmark and magnetic field were aligned. This way you can pre-empt this criticism of your magnetic conditions being not ideal and noise on the left handside of the spectrum measured (which is not uncommon).

Thank you for this helpful suggestion. We have now cited Engels et al. (2014, Nature) in this section and expanded the text to explain why electromagnetic noise at the experimental site is relevant for interpreting the behavioural responses. We also clarify the rationale for measuring electromagnetic noise and discuss the observed low-frequency (“left-hand side”) noise in the spectrum.

• Fig. 51: Suggest adapting Y-axes and using violin or box plots (e.g., panels A/B starting from 30 up to 50, etc.).

Thank you for this helpful suggestion. We have revised Fig. 5 accordingly by adapting the Y-axis scaling and replacing the original plots with box plots, as suggested.